# The Reception of Jeremiah in Modern Hebrew Literature

**Michael Avioz**

Department of Bible, Bar Ilan University, Ramat Gan 5290002, Israel; michael.avioz@biu.ac.il

**Abstract:** Looking at some illustrative examples of the reception of Jeremiah in modern Hebrew literature, this article explores how both the prophet and the book named after him were reworked by modern Hebrew authors and poets in the body of literary works in Hebrew that emerged during the late eighteenth and early nineteenth centuries in Europe in the wake of the Enlightenment.

**Keywords:** Jeremiah; Agnon; Bialik; reception of the Bible

## 1. Introduction

The Hebrew Bible has been a source of inspiration for Jewish authors throughout the ages, from the Pseudepigrapha, Qumran library, rabbinic corpus, and medieval works to modern Jewish literature. While Jeremiah belonged to the prophetic tradition, delivering oracles of doom and salvation as a messenger of God to Israel/Judah and the nations, the book recording his words exhibits unique features: an emphasis on the campaign against false prophets, Jeremiah's role as "a prophet to the nations" (Jer. 1:5), the laments in Jeremiah 11–20, and the biographical material in Jeremiah 26–45.[1]

Rather than disappearing after the canonization of the biblical text, prophets reappear in diverse forms and genres from the Second Temple period through to the present in literature, art, music, film, sculpture, and other fields. While in earlier periods, the Bible was "reworked" for religious purposes, it has also become a source for secular philosophers and writers in the modern era. From the nineteenth century onwards, Zionists attributed a prominent status to prophetic literature, regarding its values—such as social justice—as important for shaping social perception. The morality the prophets advocated also served as a foundation stone for the society Zionists sought to establish in the Land of Israel (Lammfromm 2021; Shapira 2004). The language of the prophets likewise attracted later writers. Seeking "a new style, with more beauty, freshness and vigour . . . they chose the language of the Prophets" (Klausner 1932, p. 2).

This paper explores the legacy of Jeremiah in modern Hebrew literature and poetry—i.e., the literature that emerged in the late eighteenth and early nineteenth centuries in Europe following the Enlightenment.[2]

Modern Hebrew literature draws on the Hebrew Bible by imitating its language, alluding to biblical narratives and verses, and adapting its forms. Modern Hebrew literature can be defined as "modern midrash," the authors filling in gaps, interpreting the text according to their own perspective, creating new content, confronting/criticizing/parodying the source, and employing it for personal or social use in light of their circumstances (Shaked 2005).[3] Amongst its most prominent trends are the demythization, re-mythization, and secularization of the sacred text (Shaked 2005).

Barzel (1963, p. 3) adduces six attitudes modern literary works display towards the Hebrew Bible: (1) Dismissal (primarily amongst militant groups); (2) actualization—the blending of sacred elements from the past with present outlooks; (3) a spiritual identification; (4) a romantic approach—an innocent approach to the text and recognition of subjective processing by the creator; (5) a classicism that focuses on interpreting the source; and (6) a modernist treatment—revising biblical motifs while breaking domain boundaries.

In the following, I shall survey some illustrative examples of the reception of Jeremiah in modern Hebrew literature. As we shall see below, the reuse of Jeremiah in modern Hebrew literature may reflect a change in attitude towards the Hebrew Bible. While a Jew living during the biblical or second Temple period typically regarded the biblical prophets as holy men or men of God, as secularization made inroads into modern Judaism, writers and thinkers began invoking the prophetic literature to critique Jewish society or for their own personal needs. In contrast to commentators, who address the biblical books in their entirety, verse by verse, modern Jewish poets and novelists sought to liberate themselves from the shackles of sacred Scripture. Motivated by a very different agenda, those who cited Jeremiah thus subjected his message to contemporary norms and values.

## 2. Jeremiah in Modern Hebrew Literature

Israeli artists and authors have adapted numerous Jeremianic verses and idioms. The title of Peretz Smolenskin's *Burial of the Donkey* is a reference to Jeremiah's oracle concerning Jehoiakim's demise: "With the burial of a donkey he shall be buried—dragged off and thrown out beyond the gates of Jerusalem" (Jer. 22:19). This 1873 novel depicts a nineteenth-century conflict between a progressive *maskil* and the Jewish community, its protagonist being buried like a donkey by the Jewish community and his wife becoming an apostate at the end (Smolenskin 1968; Pelli 1998).

Abraham Mapu (1808–1867) similarly draws on Jer. 12:9 for the title of his *Hypocrite Eagle* (1858), which describes contemporary Jewish society in Lithuania from a maskilic perspective (Mapu 1959). Agnon's 1939 *A Guest for the Night* alludes to Jer. 14:8 (Agnon 1968; Halevi-Wise 2014). Although none of these authors deal with Jeremiah the man or the book, they draw on his book to portray their own time and place.

Reflecting on a "prose era," Miron (1993) observes that modern Hebrew poetry regards itself as entitled to adopt the cloak of the prophet despite being prophecy without a god and on a mission without a transcendental sender. It imitates the prophetic messages in both form and content, poets from the Enlightenment onwards writing unpopular, surprising, and thought-provoking poems designed to shock their readers. The works of several modern philosophers and humanists are informed by the consciousness of just such prophetic mission. Haim Nachman Bialik (1873–1934) describes Asher Zvi Hirsch Ginsberg (1856–1927), better known as Ahad Ha'am, as a true prophet, for example, thereby taking sides in the debate between the latter and Herzl (Shoham 2003, pp. 111–21).

In his "In the City of Slaughter," Bialik undermines Jeremiah's oracles (Mintz 1984), assuming his prophetic role.[4] As the prophet is subject to God's word, so the poet is at the mercy of the muse (Mintz 1984, p. 142). His poem "A Spoken Word" opens with a description of a prophet with a prophecy of wrath in his mouth, sent to warn the people of the coming retribution (Hadari 2000, p. 100; cf. Fishelov 2019). Rather than being appreciated, the prophet is humiliated by his audience, his words falling on deaf ears. As the poem progresses, he is denounced as a false prophet who, rather than predicting disaster, offers a vain hope of "restoration and salvation." This inversion of the biblical paradigm—the Israelites wishing for prophecies of redemption and peace—appears to reflect the situation of the Jews in Bialik's own day: accustomed to pogroms and blood libels, they have no interest in words of consolation.

While Jeremiah also appears in Bialik's "City of Slaughter," herein he represents God's helplessness in the face of the 1903 Kishinev pogrom rather than testifying to God's greatness and righteousness. His mission is bound to fail because the people are unwilling to listen and because God is no longer a powerful hero.

Jeremiah's influence on Bialik is also evident in "I Have Not Gained the Light from the Unclaimed Property." Herein, the image of the hammer and rock (Jer 23:29) serves to convey the difficulty Bialik experienced in publishing his poems. Rather than being God's messenger, he is his own envoy:

I have not found light in unclaimed property,
It did not come to me by inheritance from my father.

> Rather, I hammered it out of my stone and rock and carved it from my heart.
> A spark hides in the depth of my heart, a little spark—but all mine.
> I did not borrow it from anyone, nor steal it,
> It is from and in me.
> Under the large hammer of sorrow my heart bursts, rock of my might,
> This spark sparked into my eyes, and from my eyes—to my rhymes.
> And from my rhymes fly into your hearts,
> In the morning light will ignite, vanish.
> My marrow and blood feed the fire. (Shoham 2003, pp. 123–24)

Later poets and writers objected to Bialik's self-identification, with Jeremiah, Jacob Lerner (1879–1918), and Zalman Shneour (1887–1959) being two prominent examples (Shmeruk 1999, pp. 278–85; Miron 1987, pp. 215–24).

Judah Leib Gordon (1830–1892) regarded the prophet as representing the rabbinic establishment. In his final poem, "King Zedekiah in Prison" (Gordon 1956; Nash 2003), he has Zedekiah ask Jeremiah: "What have I sinned?" (cf. Shapira 2004, p. 12). Later, Zedekiah accuses Jeremiah of being "A coward, with a surrendering soul, who advised us shame, slavery and discipline." Gordon's objection is to the authority the rabbis have assumed by representing themselves as the prophets' successors rather than the prophetic figure itself (Shoham 2003, p. 36). He thus takes a different line to that adopted by Spinoza (1623–1677) in his *Tractatus Theologico-Politicus*, who argues that the prophets interfered with the nation's affairs.

Referring explicitly to Jeremiah in his 1956 poem "The Root of My Soul," Gordon suggests that the Anatothite revives his soul (Barzel 2017). "The Flock of the Lord" likewise alludes to Jer. 13:17, declaring that the Jewish people have reached a dead end, lacking even the basic skills to continue existing as a unified entity (Holtzman 2017).

The comparison between poet and prophet reaches its peak with Uri Zvi Greenberg (1896–1981). In "With My God, the Blacksmith" (Greenberg 2003, pp. 66–70), Greenberg portrays himself as a block of metal smelted by God until he becomes a prophet whose bones burn with Jeremianic fire (cf. also "Like Chapters of Prophecy"). Like Jeremiah, who seeks to escape from God but acknowledges that he cannot, Greenberg laments: "This is my just lot." As numerous scholars note, Greenberg's awareness of his mission was genuine, the poet volunteering for the role rather than waiting for God to appoint him. He was thus known in his lifetime as "the Jeremiah of our generation" (Abrahamson 2010, p. 7). The image of God as a blacksmith, which Greenberg employs defiantly, alludes to a Yom Kippur liturgical poem that depicts man as "clay in God's hand" (Jer. 18:6) (Greenberg 2003, pp. 58–70; cf. Stahl 2021). Like Hephaestus, Greenberg's deity is thus a deformed god who produces beautiful art.

## 3. Jeremiah in Agnon's Works

One of the most well-known Jewish novelists of recent generations, Shmuel Yosef Agnon (1888–1970), seamlessly blends together biblical, talmudic, medieval, and modern Hebrew sources.[5] In "Edo and Enam," he puts the words "I want to eat *kavanim*" in the mouth of Gemulah, a member of a remote tribe (Agnon 1966, p. 178). According to Jer. 7:18, 44:7–19, these were small cakes made for Ishtar, the Queen of Heaven. Herein, Agnon Judaizes Gemulah, also revealing the process whereby an idolatrous myth is transformed into a Jewish aggada and the significance of the oscillation between these historical strata.

In *The Bridal Canopy* (Agnon 1967, p. 63), Agnon cites several Jeremianic verses: "For behold as the clay in the potter's hand" (Jer. 18:6) (Agnon 1995, p. 24); "my dear son, Ephraim" (Jer. 31:19) (p. 267). In a dialogue between a mouse and cock, he has the two animals debate the nature of human faith. Both cite biblical verses, including lines from Jeremiah:

> There was a cock that lived with a Jew. He made an easy living and lacked for
> nothing. Nonetheless he was troubled and worried and never a smile would you

catch on his face. When the month of Ellul came round at the end of the summer his troubles were doubled and he'd never crow without bursting into tears. Now a mouse lived there as well. The mouse asks the cock, Choicest of poultry, why dost thou sorrow so? If it be by reason of thy sustenance, 'tis always awaiting thee; and if it be thy dwelling, thou dwellest with human beings; yet despite all this thou 'rt grieved and terrified and quivering and crestfallen like to a helpless and weary cock. Said he, Hath not Jeremiah said, "Curst be the cock that trusteth in man," while Elihu hath told Job, "Is there an angel over him, a single counselor, one among a thousand, to tell his uprightness to Man?"; all the good things of thy speech are as nought to me when I see the master of the house taking his prayer book in hand. And why? By reason of a certain prayer, in the Order of Prayers, called "Sons of Man"; when he readeth this prayer on the appointed Eve of Atonement he taketh a cock, whirleth it about his head, saith, This cock shall go to death, and handeth it over to the slaughterer. Of me did Jeremiah lament, "I am the cock that have seen affliction."

In *And the Crooked Shall be Made Straight* (Agnon 1953, p. 88), Agnon alludes to a passage in Baba Bathra 9b in which the Sages understand Jer. 18:23: "What is the meaning of that which is written: 'Let them be made to stumble before You; deal thus with them in the time of Your anger' (Jer. 18:23)? The prophet Jeremiah said before the Holy One, Blessed be He: Master of the Universe, even when those wicked men who pursued me subdue their inclinations and seek to perform acts of charity before You, cause them to stumble upon dishonest people who are not deserving of charity, so that they will not receive reward for coming to their assistance".

In *Only Yesterday* (Agnon 2000, pp. 241, 243), he refers to Zedekiah's cave:

Sometimes they walked around the Old City walls and its seven gates, and sometimes they left from Damascus Gate and went to the Cave of Zedekiah, where King Zedekiah fled from the Chaldeans, and the cave goes underground all the way to Jericho. And opposite the Cave of Zedekiah you see the yard of the dungeon where the Prophet Jeremiah was imprisoned and the cistern where Jeremiah was thrown and the rock Jeremiah sat on and lamented the Destruction.

In *A Guest for the Night* (Halevi-Wise 2014), Agnon follows Jeremiah and Lamentations in adducing the effects of destruction in order to demonstrate the need for social renewal and reconstruction in the Land of Israel. Herein, he sets forth the Zionist ethos of homecoming after exile, the novel revolving around a married man who, after making aliyah with his family, returns to his birth town in Galicia after their new home in Jerusalem is destroyed in the Arab riots of 1929:

A verse came to my lips: "She has become as a widow." When Jeremiah saw the destruction of the First Temple, he sat down and wrote the Book of Lamentations, and he was not content with all the lamentations he wrote until he had compared the congregation of Israel to a widow and said, "She has become as a widow"— not a true widow, but like a woman whose husband has gone overseas and intends to return to her. When we come to lament this latest destruction we do not say enough if we say, "She has become as a widow," but a true widow, without the word of comparison. (Agnon 1968, p. 231)

Modern Hebrew writers also laud Gedaliah, son of Ahikam. Several writers published stories relating to this biblical figure in the 1930s and '40s, possibly reflecting the way in which the rise of the Nazis recalled the Babylonian destruction. Menachem Zalman Wolfowski's (1893–1975) trilogy *King in* Judah (1936–1937) is comprised of *Johanan son of Kareah*, *City Under Siege*, and *Last Firebrands* (Wolfowski 1964). Herein, he describes the final days of the First Temple and its destruction. The humble, brave, and loyal hero is confronted by the wicked, devious, and murderous Ishmael, son of Nethaniah, their hostility deriving from the fact that they are both in love with King Zedekiah's daughter (the princess naturally preferring the chivalrous Johanan).

Ishmael betrays Jerusalem, handing the city over to the Babylonians in anticipation of being appointed its ruler. Although the Babylonians renege on their promise, he escapes and, full of revenge, returns to murder Gedaliah (a minor figure in the story) and kidnaps his beloved. Johanan holds him off, but while, as in the biblical story, he goes down to Egypt with the remnant, he eventually returns to Israel with his wife to work the land in line with Wolfowski's Zionist-socialist principles (Eshed n.d.; Shaked 1977, pp. 425–29).

## 4. Jeremiah in the Works of Women Poets

Scholars point to the absence of Jewish women poets before the 1920s. Some explain that there were sociological and historical reasons for this inequality (See Rattok 1999; Olmert 2012, p. 50). Yet Feiner and Cohen (2006) claimed that there were women poets and novel writers during the period of the Haskala.

Rachel Bluwstein-Sela (1890–1931), or just "Rachel," is the "founding mother" of modern Hebrew poetry by women. After immigrating from Russia and finding her place in the Land of Israel, she has always read the Hebrew Bible and taught it to others (Milstein 1993). Rachel does not see any religious meaning in the biblical characters but rather uses them to describe universal feelings and emotions (Shaked 2005, pp. 211–22). She alludes to the Hebrew Bible in several poems. In one of these poems, named "Rachel," she echoes the biblical narrative of Rachel, Jacob's wife, and Jeremiah. He alludes to Jeremiah's salvation oracle in 31:14–16, which speaks of Rachel weeping in her song "El Artzi" ("To my Country"; Bluwstein 1935).

The poetess Zelda (Zelda Schneurson Mishkovsky [1914–1984]) was an observant Orthodox poet well-known among critics. The daughter and granddaughter of prominent Hasidic rabbis from the Habad dynasty (see Bar-Yosef 2007), she alludes to numerous biblical characters—Abraham, Joseph, Mephiboshet, Jonah, and Saul. Jeremiah barely makes an appearance in her work, however. In one poem, "Ba-gilgul ha-aharon" ("The Last Incarnation"), she speaks of Jeremiah as the "angel of Anatoth": "I forgot that even the prophet cried, even the angel from Anatoth cried" (translation: Falk 2004). The prophet cried for his loneliness, and perhaps the poet reflects her own pain for the lack of normal family life.

In "At the Turn of Childhood—A New Fruit" there is an explicit reference to Jeremiah's lament in Jer. 20:7:

> Slowly slowly I befriended the heavens and began to distinguish darkness from darkness night from night and I said in my heart: the name of the green nights transparent as the sea—"You enticed me, O Lord, and I was enticed" or Jeremiah's crown, and the name of the blue nights, starlight nights. (Ocker 2006)

Lea Goldberg (1911–1970) is another famous poetess. In her poems, she mostly alludes to the Song of Songs, the Book of Genesis, and Ecclesiastes (Shacham 2013). In her diaries, she recounts that her Bible teacher helped her memorize swathes of the Hebrew Bible (Aharoni and Aharoni 2005, p. 133). Nonetheless, one cannot find a significant amount of allusions to or citations from Jeremiah. She alludes to Jer. 2:2 in her poem "I loved my master." Jeremiah prophesied: "I remember the devotion of your youth, your love as a bride, how you followed me in the wilderness, in a land not sown." Goldberg (1973, 2:197–98) writes: "He has remembered the devotion of my youth and my following him in the wilderness". In one of her last songs, "the world is heavy on our eyelids" Goldberg uses the phrase "The day turned" (*hayom pana*), which originates in Jer. 6:4 (Back 2017, p. 145).

## 5. Conclusions

This paper reviews the reuse of Jeremiah in modern Hebrew literature. One of the reasons this neglects Jeremiah and his prophecies in comparison to other biblical figures is likely a function of the fact that while biblical narrative served as a literary goldmine, prophecy was a far more cryptic matter. Biblical stories address issues such as love, hate, jealousy, sex scandals, and upheavals; prophecy—and Jeremiah's oracles in particular—is exacting, agonizing, and full of despair. During the *Haskala*, writers only embraced the

biblical style to satisfy their (literary/personal) needs (e.g., Mapu), with few calling for any practical observance or espousing biblical values. Those who did reuse Jeremiah drew expressions, verses, and concepts from the text, adapting it to their present-day reality.

Gordon and Wolfowski use Jeremiah more than Bialik and Agnon, the latter only adducing certain expressions or motifs. Agnon's neglect of Jeremiah may reflect his break with tradition (Pardes 2013). Since then, the Hebrew Bible has been subject to increasing secularization in modern Hebrew literature, evinced in the most recent reworkings of Jeremiah, wherein Jeremiah assumes a parodic, dystopian, satirical guise (Tammuz 1984; Burstein 2016).[6]

**Funding:** This research received no external funding.

**Institutional Review Board Statement:** Not applicable.

**Informed Consent Statement:** Not applicable.

**Data Availability Statement:** Not applicable.

**Conflicts of Interest:** The author declares no conflict of interest.

## Notes

[1]   For other ways in which Jeremiah is unique among the prophets, see (Brown 2010, pp. 25–26).

[2]   Studies of the reception of the Book of Jeremiah have largely overlooked this aspect: see (Bogaert 1997; Najman and Schmid 2016; Lundbom et al. 2018; Fischer 2016; Stulman and Silver 2021). For anthologies of compositions that appeal to the Hebrew Bible, see (Elkoshi 1953; Rabinowitz and Yardeni 1962–1963; Shaked 2005). Callaway (2020) does mention Gordon's poem. See below.

[3]   Needless to say, readers of these poems and novels must have full command of the Hebrew Bible in order to be aware of the allusions, rewriting, or subversion of the biblical verses.

[4]   Scholars debate the precise meaning of this poem: see (Raz 2013) and the bibliography cited therein.

[5]   My thanks go to Prof. Hillel Weiss of Bar-Ilan University for assistance in locating the citations from Agnon's works. Other brief mentions of Jeremiah in Agnon's novels are: *Esterlein Yekirati: Michtavim* (*My Dear Esterlein: Letters*; Agnon 1983, p. 68); *Present at Sinai* (Agnon 1994); Sefer, sofer ve-sippur ("Book, Writer, and Story"; Agnon 1978, p. 36); *Ha'esh veha'etzim* ("The Fire and the Wood" Agnon 1971, p. 159).

[6]   See (Yosef-Paz 2018). Episode 15 of season 4 of the satirical TV show "Ha-yehudim baim" ("The Jews are Coming") aired on 8 September 2020, relates to Jeremiah. See also Yehoshua (2009).

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
