# Peer review of "The Reception of Jeremiah in Modern Hebrew Literature"

_religions, doi:10.3390/rel13030215_

Round 1
Reviewer 1 Report
Please see attached document.

Reviewer 2 Report
I do not know if the editors of this special issue requested this short length of article. In its present form it is definitely useful and publishable “as is.” I find it interesting and coherent. However, If the author would like to expand this, now, or in the future, I offer the following suggestions:
The essay’s overall design, questions, hypotheses and methods could be more compellingly highlighted by letting us know why this survey is interesting/useful/important?
The arguments surrounding and linking the different parts of this survey could be presented in a more compelling fashion by foregrounding commonalities and differences among the different examples, and/or by entering into conversation with Hillel Barzel’s six attitudes toward Biblical Citations (listed on line 37ff): the surveyed examples course be organized and analyzed according to those attitudes; or the attitudes themselves can be subjected to a critique. Do all examples fit into these six attitudes? Do they overlap? Are some more robustly present than others? Are other attitudes toward biblical citations in regards to Jeremiah not noticed by Barzel? This could be done in a friendly conversational tone rather than as a competition.
The conclusion/results likewise can be articulated more boldly to highlight why the survey is useful/interesting/important?
Overall, this survey contributes substantially to scholarship on modern Hebrew literary engagements with Prophetic Scriptures because it presents a comprehensive survey of allusions to Jeremiah in that corpus. However, the article does not engage in a substantial conversation with the scholars cited nor does it offer any argument about the material that it surveyed. As such, it is encyclopaedic rather than analytic; a useful scholarly intervention that provides an excellent basis for expansion into more substantial engagement with the examples, as well as with the secondary scholarship on the topic and key issues emerging herein.
It is very well written, with hardly any need for copyediting. I note a few tiny issues that I spotted:
line 19 remove for
line 209 lowercase Rachel Does
When citing Shaked parenthetically, distinguish whether it is Malka or Gershon.
line 222 articulate what is meant by “lack of normal family life” for Zelda. (Or, better still, contextualize in relation to other references in Zelda’s work to loneliness, rather than simply extrapolating on her life and childlessness.)
Overall, thank you. I enjoyed and profited from this survey!
Reviewer 3 Report
This is well researched and clearly written. But beyond surveying the limited appearances of Jeremy (both objectively and in contrast to the appearance of other prophets) in a modern Hebrew authors, the article poses no significant questions that would interest a scholar either of literature or, especially for this Journal, of religion: What does the afterlife of prophetic writing in modern Hebrew literature teach us about the construction of modern Israeli culture? Of modern Judaism or Jewish culture? Why do the particular writers who reference Jeremiah use him, and do others not? Why does Jeremiah get less use than other prophets? There is surely a range of questions that the raw data gathered here can answer. But the details of the appearances alone, which is all that is presented here, don't go far enough in creating an article of full scholarly value.
Round 2
Reviewer 3 Report
The several added paragraphs help a lot in suggesting the meaning and importance of the cited examples. They accomplish what was needed, turning a list of examples into a statement of a point regarding the distinctive use of biblical materials—Jeremiah in particular—in modern Hebrew literature. Only issues I see at this point are some minor typos and very small problems with English usage in the revisions.